# Low serum brain-derived neurotrophic factor may predict poor response to cardiac rehabilitation in patients with cardiovascular disease

Yuya Tsukada[1,2]*, Yasuhiro Nishiyama[3], Michiya Kishimoto[1], Takeshi Nago[1], Haruhito Harada[3], Hiroshi Niiyama[3], Atsushi Katoh[3], Hiroo Matsuse[4], Hisashi Kai[3]

**1** Division of Rehabilitation, Kurume University Medical Center, Kurume, Fukuoka, Japan, **2** Department of Physical Therapy, Miyazaki Medical Association Hospital, Miyazaki, Japan, **3** Department of Cardiology, Kurume University Medical Center, Kurume, Fukuoka, Japan, **4** Division of Rehabilitation, Kurume University Hospital, Kurume, Fukuoka, Japan

☯ These authors contributed equally to this work.
* egaodeikouyo2525@gmail.com

**Data Availability Statement:** All relevant data are within the manuscript and its Supporting Information files.

## Abstract

### Background

It has been shown that serum brain-derived neurotrophic factor (BDNF) is associated with skeletal muscle energy metabolism and that BDNF is a predictor of mortality in heart failure patients. However, little is known about the relationship between BDNF and cardiac rehabilitation (CR). Therefore, this study retrospectively investigated the effects of baseline serum BDNF levels on the CR-induced exercise capacity improvement in patients with cardiovascular disease (CVD).

### Methods

We assigned 99 CVD patients (mean age 71±12 years, male = 60) to Low, Middle, and High groups based on the tertiles of baseline BDNF levels. Cardiopulmonary exercise testing was done using supervised bicycle ergometer twice before and after 3 weeks of CR. Analysis of covariance (ANCOVA) followed by post-hoc analysis using Tukey's HSD test was conducted to assess the multivariate associations between baseline BDNF levels categorized by BDNF tertiles (as independent variable) and %increases in AT and peak $VO_2$ after 3-week CR (as dependent variables) after adjustment for age and gender (as covariates), as a main statistical analysis of the present study.

### Results

The higher the baseline BDNF levels, the better nutritional status evaluated by the CONUT score (p<0.0001). Baseline anaerobic threshold (AT) and peak oxygen uptake (peak $VO_2$) were similar among the three groups. ANCOVA followed by post-hoc analysis revealed that age- and gender-adjusted %increases in peak $VO_2$ after 3-week CR were positively associated with baseline BDNF levels (p = 0.0239) and Low BDNF group showed significantly lower %increase in peak $VO_2$ than High BDNF group (p = 0.0197). Significant association was not found between baseline BDNF and %increase in AT (p = 0.1379).

**Funding:** "Research grants were received from the Japanese Circulation Society. Additionally, this work was partially supported by JSPS KAKENHI Grants (HK, 19K08503)." The funders had no role in study design, data collection and analysis, decision to publish, or preparation of the manuscript.

**Competing interests:** The authors have declared that no competing interests exist.

## Conclusions

Low baseline BDNF levels were associated with malnutrition in CVD patients. A positive association between baseline BDNF levels and CR-induced increases in peak $VO_2$ was found. It was suggested that CVD patients with low baseline BDNF levels may be poor responders to CR.

## Introduction

Exercise-based cardiac rehabilitation (CR) increases aerobic capacity and exercise tolerance, and improves prognosis in patients with cardiovascular disease (CVD) [1,2]. Cochrane reviews have reported on the beneficial effects of CR on quality of life, morbidity and mortality [3]. However, several other studies reported that 21–23% of patients (non-responders) failed to show a favorable response to this training [4,5]. Therefore, it is necessary to identify non-responders to help in the early implementation of exercise prescriptions optimized to such individuals. Several non-modifiable factors (sex, age, comorbidities, etc.) and modifiable factors (baseline physical fitness, exercise type, dose, etc.) have been reported to be predictors of response to exercise training [6].

Brain-derived neurotrophic factor (BDNF) is a member of the neurotrophin family, which regulates neuronal differentiation, maintenance and survival in central nervous systems [7,8]. It has been shown that low levels of BDNF are associated with depression and cognition disorders [9–11]. Since exercise produces BDNF in skeletal muscle, it is thought that BDNF is associated with the function of myokines. In heart failure, reduced exercise capacity is due to peripheral skeletal muscle abnormalities [12]. A previous study has also shown that low levels of BDNF are related to impaired exercise capacity and poor prognosis in patients with chronic heart failure [13]. However, the relationship with baseline BDNF levels and response to CR in patients with CVD has yet to be definitively established. Therefore, we investigated the effects of baseline serum BDNF levels on the response to CR before and after 3 weeks of such training in CVD patients.

## Methods

### Study patients

We retrospectively studied 99 consecutive patients with CVD (age: 71±12, males 60) who were admitted to Kurume University Medical Center from February 2013 to April 2015 and underwent exercise stress test and serum BDNF sampling before and after 3 weeks of CR. Patients with severe renal failure, severe hepatic disease and malignant neoplasms were excluded from the study. All patients received guideline-directed medical therapy. Medications were not changed during the study period. Written informed consent was obtained from all patients, and the study was approved by the Ethics Committee of Kurume University (approval number: 18391).

### Blood samples

Blood samples were collected in the morning following an overnight fast. After resting in a supine position for at least 20 min, the samples were taken from the antecubital vein, and after immediate centrifugation, aliquots were stored at –70˚C until analysis. BDNF concentrations were quantitatively measured by enzyme-linked immunosorbent assay (ELISA) using the

Quantikine® ELISA Human Free BDNF Immunoassay from R&D Systems (Minneapolis, MN, USA).

## Nutritional assessments

We assessed the nutritional status at admission using the Controlling Nutritional Status (CONUT) scoring system. As previously reported, the CONUT score is determined by serum albumin, total cholesterol concentrations and total peripheral lymphocyte counts [14].

## Muscle mass measurements

Muscle mass was measured by a bioelectrical impedance assay using the In Body S10 body composition analyzer (Biospace, Tokyo, Japan). The conditions for bioelectrical impedance assay measurements were: (i) fasting for 4 hours before measurements; (ii) bladder voided before measurements; and (iii) no exercise in the 8-hour period prior to measurements [15]. Edematous patients were examined only after an improvement in their edema. Absolute appendicular muscle mass was calculated as the sum of the muscle mass of the arms and legs. Skeletal muscle mass index (SMI ($kg/m^2$)) was then calculated by dividing the appendicular muscle mass by the square of the height in meters.

## Exercise stress test and cardiac rehabilitation

All of the 99 patients underwent a symptom-limited exercise stress test before and after the exercise training. Exercise stress tests were performed using an electrically braked bicycle ergometer (BE250, Fukuda Denshi, Tokyo, Japan). A ramp protocol, starting at a workload of 20 watts and that was incrementally increased by 1 watt every 6 seconds (10 watt/min) was used. ECG and SBP were recorded every 1 minute before, during and after exercise (ML-9000, Fukuda Denshi, Tokyo, Japan). Criteria for stopping the exercise test included life-threatening arrhythmias, ST-segment depression or elevation >0.2mV, a fall in the SBP >20mmHg, or attaining the predicted target heart rate (HR) [(220 –age) x 0.85]. In all patients, expired gas was collected and analyzed continuously with an AE-310S gas analyzer (Minato Co., Osaka, Japan). Peak oxygen uptake ($VO_2$) was defined as the highest $VO_2$ value achieved at peak exercise. The anaerobic threshold (AT) was defined using the V-slope method.

Comprehensive CR was started on admission to the hospital. Comprehensive CR was defined as a tailored multidisciplinary intervention that included clinical evaluation, management and modification of cardiovascular risk factors, physical activity counseling, prescription of an appropriate exercise training program, dietary counseling, as well as psychological, social, and vocational support [16]. Exercise training was started from an early stage of admission, except in those patients with an unstable condition such as acute coronary syndrome, congestive heart failure and life-threatening arrhythmia. Forty minutes of supervised bicycle ergometer training was performed 5 days/week for 3 weeks in the hospital. Exercise intensity was set at the AT level, which was determined by the entry exercise test. Adherence to the inpatient CR program was almost complete for all study subjects.

## Statistical analysis

Values are presented as mean ± standard deviation or median (95% confidence interval). The Wilk-Shapiro test was used to assess the normality of distribution of the data. For clinical characteristics, comparisons between groups for continuous variables were performed using ANOVA with post hoc pairwise comparisons, unpaired two-sample t-tests or the Wilcoxon signed rank test, as appropriate. The comparison between before and after CR was performed

using a paired t-test. Analysis of covariance (ANCOVA) was conducted to assess the multivariate associations between baseline BDNF levels (as independent variable) and %increases in AT and peak $VO_2$ after 3-week CR (as dependent variables) after adjustment for age and gender (as covariates), as a main statistical analysis of the present study. And, ANCOVA followed by post-hoc analysis using Tukey's HSD test was used to compare age- and gender-adjusted % increases in AT and peak $VO_2$ after 3-week CR. Baseline BDNF levels were treated as a categorical data based on BDNF tertiles. Age and gender were treated as a continuous and categorical data, respectively. All statistical analyses were performed using a JMP 14.0.0 statistical software (SAS Institute Inc., Cary, NC, USA). A probability value of <0.05 was considered to be significant.

## Results

### Baseline characteristics

Ninety-nine CVD patients were allocated to the Low, Middle and High groups based on the tertiles of baseline BDNF levels. Table 1 shows the baseline characteristics of the enrolled patients. The CONUT score decreased and hemoglobin increased significantly, as BDNF levels increased, respectively. There were no significant differences among the groups for age, sex, main disease, comorbidities, medication, ejection fraction, or SMI.

### Exercise stress test parameters before and after exercise training

AT and peak $VO_2$ were not different among Low, Middle, and High BDNF groups before exercise training. Fig 1 shows the effect of exercise on AT and peak $VO_2$ in the three groups. The 3-week exercise training increased the AT (from 10.5±2.9 to 11.4±3.6, p<0.01) and peak $VO_2$ (from 14.5±4.3 to 15.7±4.5, p<0.05) in High BDNF group and increased the AT (from 10.4 ±2.2 to 11.4±2.6, p<0.05) in Middle BDNF group. In contrast, exercise training did not cause any changes in the AT and peak $VO_2$ in Low BDNF group. As shown in Fig 2, ANCOVA followed by post-hos analysis showed that the age- and gender-adjusted %increases in peak $VO_2$ after 3-week CR were positively associated with baseline BDNF levels (p = 0.0239) and that High BDNF group had significantly higher %increase in peak $VO_2$ than Low BDNF group (p = 0.0197). In contrast, significant association between baseline BDNF and %increase in AT after 3-week CR was not found (p = 0.1379). When the age- and gender-adjusted %increases in AT were compared between Low BDNF group and Middle+High BDNF group, the adjusted %increase in AT tended to be higher in Middle+High BDNF group (9.53±2.16%) than in Low BDNF group (2.51±3.17%) although the difference was not significant (p = 0.0689).

## Discussion

The present study demonstrated that BDNF level was negatively associated with the CONUT score in CVD patients. The 3-week CR improved both the AT and peak $VO_2$ in High BDNF group, whereas neither AT nor peak $VO_2$ were changed in Low BDNF group. After adjustment for age- and gender, a positive association between baseline BDNF levels and %increases in peak $VO_2$, but not in AT, was found after 3-week CR, and Low BDNF group had significantly lower %increase in peak $VO_2$ than High BDNF group.

Early identification of non-responders could be helpful for CR, as this would allow for the tailoring and optimization of exercise prescriptions for individual CVD patients. The function of the lungs, heart, peripheral circulation, pulmonary circulation, and peripheral skeletal muscle may play an important role in the determinants of exercise tolerance [17]. Many previous reports have suggested that peripheral function, mainly of the skeletal muscle, is more

**Table 1. Baseline characteristics and demographics.**

| | | Low BDNF | Middle BDNF | High BDNF |
|---|---|---|---|---|
| | | (n = 33) | (n = 33) | (n = 33) |
| Serum BDNF (pg/ml) | | 12100(10631–13270) | 19600(1894–20107) | 26400(25833–29736) |
| Age | | 77(27–91) | 73(55–87) | 70(35–87) |
| Sex (M/F) | | 11/22 | 18/15 | 20/13 |
| Main Disease | | | | |
| | Coronary artery disease | 9 | 15 | 13 |
| | Heart failure | 14 | 6 | 14 |
| | Arrhythmia | 2 | 6 | 3 |
| | Hypertension | 0 | 3 | 2 |
| | Arteriosclerosis obliterans | 1 | 0 | 1 |
| | Post operated | 5 | 1 | 0 |
| | Others | 2 | 2 | 0 |
| Comorbidities (%) | | | | |
| | hypertension | 17 | 20 | 23 |
| | diabetes mellitus | 11 | 15 | 11 |
| | hyperlipidaemia | 12 | 11 | 11 |
| | hyperuricemia | 3 | 4 | 4 |
| Drug (%) | | | | |
| | ARB | 45.5 | 45.5 | 39.4 |
| | ACE inhibitor | 24.2 | 12.1 | 27.3 |
| | β blocker | 60.2 | 51.5 | 52.5 |
| | Ca blocker | 18.2 | 42.4 | 30.3 |
| | Statin | 45.5 | 45.5 | 39.4 |
| CONUT | | 3(2–4) | 2(1–3)* | 1(1–2) [†] |
| Hemoglobin (g/dL) | | 10.9±2 | 12.8±1.9* | 13.4±1.5[†] |
| LVEF (%) | | 58(51–61) | 64(58–66) | 62(52–61) |
| SMI (kg/m$^2$) | | 6.6±1.3 | 6.8±1.6 | 7.5±1.4 |

Notes: Data were expressed as mean

± standard deviation or as median (95% confidence interval)

* $p < 0.01$ between low and middle BDNF group

[†] $p < 0.01$ between low and high BDNF group; CONUT, controlling nutritional status; LVEF, left ventricular ejection fraction; SMI, skeletal muscle mass index.

important than central circulation, which includes cardiac function, for exercise tolerance in heart failure patients [12,18–20]. Recent studies have demonstrated that achieved peak VO$_2$ after CR can serve as a predictor of long-term survival in patients with CVD [21,22]. Therefore, this suggests that more attention needs to be paid to skeletal muscle function and quality in order to determine CVD and heart failure patient responses to CR [23]. BDNF is known to have a myokine regulating function and be involved in the maintenance of skeletal muscle [24]. Therefore, we hypothesized that serum BDNF could be used as a predictor of the response to CR in CVD patients.

In previous studies, poor response to CR was attributed to age, gender, presence of diabetes, and exercise tolerance at the time of initiation [5,6]. In the present study, the prevalence of these factors was similar among Low, Middle and High BDNF groups, as well as SMI, left ventricular ejection fraction, and the prevalence of heart failure, coronary artery disease, other cardiovascular diseases, the comorbidities, and medications. It is noteworthy that baseline AT and peak VO$_2$, parameters of anaerobic capacity, did not differ among the three groups and

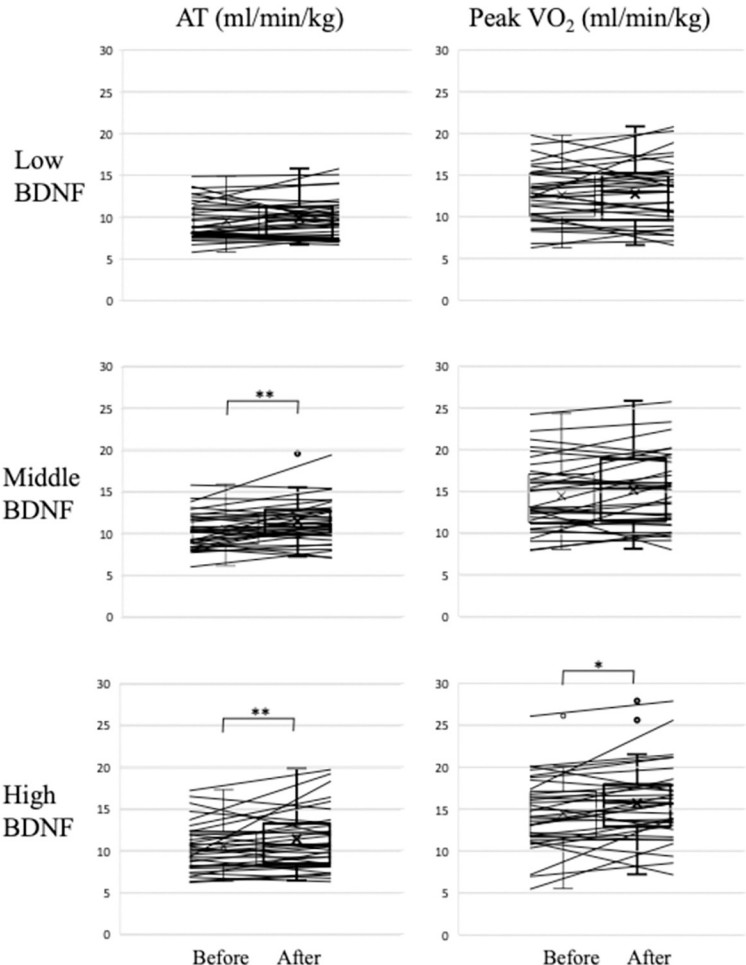

**Fig 1. Effects of 3-week cardiac rehabilitation on anaerobic capacity in CVD patients.** AT, anaerobic threshold. *p<0.05 and **p<0.01 vs. before rehabilitation.

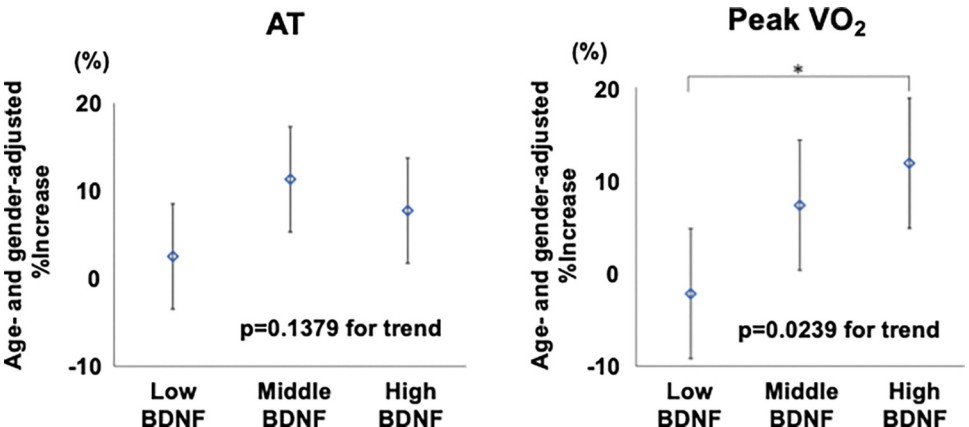

**Fig 2. Age- and gender-adjusted %increase in AT and peak VO$_2$ after 3-week cardiac rehabilitation in CVD patients.** Bar = 1xSD *p<0.05 vs. High BDNF group.

that even after adjustment for age and gender, the CR-induced increase in peak $VO_2$ was significantly lower in Low BDNF group than in High BDNF group. Thus, it is possible that low baseline BDNF may be an independent predictor of non-responders for CR in CVD patients.

BDNF was originally discovered in the brain and determined to be a member of the neurotrophin family, which regulates various neurotrophic functions including neuroregeneration, neuroprotection, and synaptic plasticity [8]. BDNF is produced in skeletal muscle cells in response to contraction and is associated with enhanced fat oxidation, with serum BDNF levels increased by exercise training [25,26]. Fukushima *et al.* showed that serum BDNF was significantly positively correlated with exercise tolerance in patients with heart failure [13]. Matsumoto *et al.* reported that BDNF administration in mice with myocardial infarction improved skeletal muscle mitochondrial dysfunction and exercise capacity [27]. These findings suggest that serum BDNF may reflect skeletal muscle mitochondrial function. Fukushima et al. have shown a close association serum BDNF and peak oxygen uptake in patients with heart failure [13,28]. In the present study, peak $VO_2$ did not differ among Low, Middle, and High BDNF groups before exercise training (p = 0.1113), although there was a marginal association between the baseline serum BDNF and peak $VO_2$ in the whole patients enrolled (r = 0.2116, p = 0.0355). The reason of the differences between the previous and present studies was unknown. One of the possible explanations was that more than half of patients with CVD other than heart failure were included in this study. Furthermore, our present results showed that there was no difference in the SMI among Low, Middle, and High BDNF groups. It has been suggested that while BDNF is associated with skeletal muscle function and/or energy metabolism [25], it is not connected with skeletal muscle mass in CVD patients. Therefore, the lack of improvement in exercise tolerance in Low BDNF group may be related to abnormality of skeletal muscle function and metabolism, such as mitochondrial dysfunction. Nakano et al. reported that serum BDNF levels were associated with skeletal muscle strength, but not with skeletal muscle mass, in patients with heart failure [29]. From this study, the association between serum BDNF and skeletal muscle strength remains undetermined in CVD patients. This issue should be addressed in future study.

Also of note is that as compared to Middle and High BDNF groups, there was a significantly higher CONUT score and lower hemoglobin in Low BDNF group, which suggests that the nutrition state was poorer in Low BDNF group. Malnutrition causes fatigue in skeletal muscle as well as a loss of skeletal muscle mass [30]. Hirabayashi *et al.* reported that malnutrition impaired the metabolic capacity in both fast and slow muscles via AMPK-independent SIRT1 inhibition induced by increased oxidative stress in a malnutrition rat model [31]. Furthermore, Kootaka *et al.* demonstrated that malnutrition was associated with a low physical function and increased mortality risk in patients with CVD [32]. When taken together with our current findings, these results suggest that it is possible that malnutrition induces impaired skeletal muscle BDNF production and reduced serum BDNF levels. Thus, this could subsequently lead to skeletal muscle mitochondrial dysfunction along with a resultant poor response to CR in CVD patients.

Montero *et al.* showed that individual non-response to exercise training was abolished by increasing the dose and duration of the training programs in healthy volunteers [33]. Thus, this suggests that it might be possible to adjust the training program in order to avoid poor responses to CR if serum BDNF levels are low at the time of CR initiation. This would be very helpful for tailoring and optimizing exercise prescriptions in CVD patients.

These were several limitations for the present study. First, CR in this study was performed during a 3-week period in hospitalized CVD patients during the subacute phase. Therefore, it remains unknown whether serum BDNF can be used to predict the responses to longer exercise training during the chronic phase of CVD. Second, all patients in our study performed an

exercise stress test, and had a physical activity above a certain defined level. Thus, it remains unknown if there is an association between BDNF levels and the response to CR in patients with a lower physical activity. Next, it is interesting to know serum BDNF change after exercise treatment. Exercise-induced BDNF change and the association between the BDNF and the effectiveness of CR should be determined in future study. From the present study, it remains unknown why significant association between baseline BDNF and the CR-induced increase in peak VO$_2$, but not in AT, was found. A possible explanation of this discrepancy could be that AT changes are less clearly manifested than peak VO$_2$ changes because the effects of BDNF levels on AT would involve more complex and multifactorial mechanism compared with the effects on peak VO$_2$. Namely, AT serves as a comprehensive measure of exercise tolerance including the respiratory, circulatory, and metabolic processes associated with exercise, whereas peak VO$_2$ is an indicator of exercise tolerance based primarily on oxygen delivery capacity [34]. Another possibility is that the 3-week CR period during hospitalization may be too short for baseline BDNF levels to affect AT changes. Molecular and physiological mechanisms whereby BDNF affects peak VO$_2$ and AT should be determined in future studies. Finally, we only evaluated a small number of patients in the present study. A further investigation using a large number of patients and a longer period of CR will need to be undertaken in order to provide a more definitive conclusion.

In conclusion, low baseline BDNF levels were associated with malnutrition in CVD patients. And, a positive association between baseline BDNF levels and the CR-induced increases in peak VO$_2$ was shown. It was suggested that CVD patients with low baseline BDNF levels may be poor responders to CR during the subacute phase. Therefore, BDNF can be a helpful biomarker for creating a tailored, optimized exercise program in CVD patients. However, this study is a small study population and therefore can be only hypothesis-generating at this stage.

## Supporting information

**S1 File.**
(XLSX)

## Author Contributions

**Data curation:** Michiya Kishimoto, Hiroshi Niiyama, Atsushi Katoh.

**Formal analysis:** Haruhito Harada.

**Investigation:** Takeshi Nago, Hiroshi Niiyama, Atsushi Katoh.

**Supervision:** Takeshi Nago, Hiroo Matsuse, Hisashi Kai.

**Validation:** Haruhito Harada, Hiroo Matsuse, Hisashi Kai.

**Visualization:** Hisashi Kai.

**Writing – original draft:** Yuya Tsukada, Yasuhiro Nishiyama, Hisashi Kai.

**Writing – review & editing:** Yuya Tsukada, Yasuhiro Nishiyama, Hiroo Matsuse, Hisashi Kai.

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
