## [Decision Letter · Decision Letter 0]

7 Feb 2023

PONE-D-23-00760Low serum Brain-Derived Neurotrophic Factor may predict poor response to cardiac rehabilitation in patients with cardiovascular diseasePLOS ONE

Dear Dr. Tsukada,

Thank you for submitting your manuscript to PLOS ONE. After careful consideration, we feel that it has merit but does not fully meet PLOS ONE’s publication criteria as it currently stands. Therefore, we invite you to submit a revised version of the manuscript that addresses the points raised during the review process.

We look forward to receiving your revised manuscript.

Kind regards,

Rabbanie Tariq Wani

Academic Editor

PLOS ONE

Journal Requirements:

- https://doi.org/10.1093/eurheartj/ehy565.P2552

In your revision ensure you cite all your sources (including your own works), and quote or rephrase any duplicated text outside the methods section. Further consideration is dependent on these concerns being addressed.

3. In the ethics statement in the Methods and online submission information, please ensure that you have specified (1) whether consent was informed and (2) what type you obtained (for instance, written or verbal, and if verbal, how it was documented and witnessed). If your study included minors, state whether you obtained consent from parents or guardians. If the need for consent was waived by the ethics committee, please include this information.

Additional Editor Comments:

One of the reviewers have suggested for a minor review, please find there comments attached

Reviewers' comments:

Reviewer's Responses to Questions

**Comments to the Author**

1. Is the manuscript technically sound, and do the data support the conclusions?

Reviewer #1: Yes

2. Has the statistical analysis been performed appropriately and rigorously? 

Reviewer #1: Yes

3. Have the authors made all data underlying the findings in their manuscript fully available?

Reviewer #1: Yes

4. Is the manuscript presented in an intelligible fashion and written in standard English?

Reviewer #1: Yes

5. Review Comments to the Author

Reviewer #1: The authors showed that serum BDNF levels can predict the effect of exercise therapy on patients with cardiovascular disease. Furthermore, they showed that serum BDNF levels were associated with patient nutritional status. This reviewer considers this study to be an interesting study. I have some questions.

1. Nakano et al. reported that serum BDNF levels were associated with skeletal muscle strength, but not skeletal muscle mass (Int Heart J 2020;61(1):96-102). Present study also showed no association between BDNF levels and skeletal muscle mass, consistent with previous result. What indicators of skeletal muscle strength are measured in this study?

2. Previous reports have shown that serum BDNF levels are decreased in patients with heart failure and that serum BDNF and peak oxygen uptake are closely related (J Card Fail 2015;21(4):300-306, Int J Cardiol 2013;168:e142-144). In this study, peak oxygen uptake before exercise therapy was not associated with serum BDNF levels. Please discuss these differences.

3. BDNF is believed to be secreted from skeletal muscle during exercise. Did serum BDNF levels change after exercise therapy?

4. FIG. 1 shows the before-and-after changes of an individual patient. Therefore, please show the graph with a line connecting the points before and after so that you can see the change.

6. PLOS authors have the option to publish the peer review history of their article (what does this mean?). If published, this will include your full peer review and any attached files.

Reviewer #1: No

---

## [Author Response · Author response to Decision Letter 0]

18 Apr 2023

Additional Editor Comments:

One of the reviewers have suggested for a minor review, please find there comments attached

Reviewers' comments:

Reviewer's Responses to Questions

Comments to the Author

1. Is the manuscript technically sound, and do the data support the conclusions?

Reviewer #1: Yes

2. Has the statistical analysis been performed appropriately and rigorously?

Reviewer #1: Yes

3. Have the authors made all data underlying the findings in their manuscript fully available?

Reviewer #1: Yes

4. Is the manuscript presented in an intelligible fashion and written in standard English?

Reviewer #1: Yes

5. Review Comments to the Author

Reviewer #1: The authors showed that serum BDNF levels can predict the effect of exercise therapy on patients with cardiovascular disease. Furthermore, they showed that serum BDNF levels were associated with patient nutritional status. This reviewer considers this study to be an interesting study. I have some questions.

1. Nakano et al. reported that serum BDNF levels were associated with skeletal muscle strength, but not skeletal muscle mass (Int Heart J 2020;61(1):96-102). Present study also showed no association between BDNF levels and skeletal muscle mass, consistent with previous result. What indicators of skeletal muscle strength are measured in this study?

Response:

Unfortunately, we have no data on skeletal muscle strength in this study. We have thus added the following sentences to the revised manuscript, to read:

Discussion (page 16, lines 205-209)

“Nakano et al. reported that serum BDNF levels in patients with heart failure were associated with skeletal muscle strength, but not with skeletal muscle mass [28]. Given that finding, the association between serum BDNF levels and skeletal muscle strength in CVD patients warrants attention. This issue should be addressed in future research.”

The new reference you so kindly provided has been added as Reference 28.

“28. Nakano I, Kinugawa S, Hori H, et al. Serum brain-derived neurotrophic factor levels are associated with skeletal muscle function but not with muscle mass in patients with heart failure. Int Heart J 2020;61:96-102.”

2. Previous reports have shown that serum BDNF levels are decreased in patients with heart failure and that serum BDNF and peak oxygen uptake are closely related (J Card Fail 2015;21(4):300-306, Int J Cardiol 2013;168:e142-144). In this study, peak oxygen uptake before exercise therapy was not associated with serum BDNF levels. Please discuss these differences. 

Response:

We greatly appreciate this pertinent comment. Discussion of this issue has been added to the revised manuscript, as follows:

Discussion (page 15, line 193 to page 16, line 200)

“Fukushima et al. showed a close association between serum BDNF and peak oxygen uptake in patients with heart failure [26,27]. In the present study, peak VO2 did not differ significantly among the Low, Middle, and High groups before exercise training (p=0.1113), although a marginal association was identified between baseline serum BDNF and peak VO2 in the total patient cohort (r=0.2116, p=0.0355). The reason for these differences between studies is unknown. One possible explanation is that more than half of the patients in the present study had CVD other than heart failure, whereas Fukushima et al. investigated the cohorts enrolling only patients with heart failure.”

The papers you mentioned have now been cited as References #26 and 27.

“26. Fukushima A, Kinugawa S, Homma T, et al. Serum brain-derived neurotropic factor level predicts adverse clinical outcomes in patients with heart failure. J Card Fail 2015;21:300-306

27. Fukushima A, Kinugawa S, Homma T, et al. Decreased serum brain-derived neurotrophic factor levels are correlated with exercise intolerance in patients with heart failure. Int J Cardiol 2013;168:e142-144.”

3. BDNF is believed to be secreted from skeletal muscle during exercise. Did serum BDNF levels change after exercise therapy?

Response:

We agree with the point you are making, but unfortunately we did not measure serum BDNF levels after exercise training. The following sentences have been added to the revised manuscript to address this:

Discussion (page 18, lines 233 to page 19, line 235)

“Next, it would be interesting to know whether serum BDNF changes after exercise treatment. Exercise-induced BDNF changes and the associations between BDNF and the effectiveness of CR should be investigated in future research.”

4. FIG. 1 shows the before-and-after changes of an individual patient. Therefore, please show the graph with a line connecting the points before and after so that you can see the change.

Response:

Thank you for this suggestion. We have added an appropriate indication of the intervention period in the revised version of Figure 1.

6. PLOS authors have the option to publish the peer review history of their article (what does this mean?). If published, this will include your full peer review and any attached files.

Do you want your identity to be public for this peer review? For information about this choice, including consent withdrawal, please see our Privacy Policy.

Reviewer #1: No

---

## [Decision Letter · Decision Letter 1]

5 Jul 2023

PONE-D-23-00760R1

Low serum brain-derived neurotrophic factor may predict poor response to cardiac rehabilitation in patients with cardiovascular disease

PLOS ONE

Dear Dr. Tsukada,

Thank you for submitting your manuscript to PLOS ONE. One additional expert reviewer carefully evaluated your manuscript. The comments are included below. Although the reviewer and I find merit in your study, I myself found concerns about the statistical analysis that precludes its acceptance for publication in PLOS ONE.

I invite you to respond to the reviewer’s comments and my own, and submit a revision if you believe you can adequately address the concerns. Please note that this request for revision does not imply that the manuscript will ultimately be accepted, but that the manuscript must be meticulously revised before further consideration is provided.

Your revised submission must include an itemized, point-by-point response to the comments of the reviewer and editor.

We look forward to receiving your revised manuscript.

Kind regards,

Yuichiro Nishida

Academic Editor

PLOS ONE

Additional Editor Comments :

Major comments:

The topic of the present study is interesting, and the results are potentially important for the development of tailored, optimized exercise programs for patients with cardiovascular disease. However, I have concerns regarding the statistical analysis. To answer the main research question, you used paired t-tests to assess pre-post changes in AT and Peak VO2 within groups. This is not sufficient to answer the research question. Instead, you should statistically compare pre-post changes in AT and Peak VO2 “between” the three groups (instead of “within” groups). For instance, the changes (increases) in AT and Peak Vo2 in the high BDNF group are significantly greater than the changes (increases) in AT and Peak Vo2 in the low BDNF group? Additionally, to adjust for potential confounding factors, an analysis of covariance (ANCOVA) may be used to calculate the adjusted means (and 95% confidence intervals) of the changes in AT and Peak VO2 in the three groups. Although there were no statistically significant differences among the three groups for age, sex, main disease, comorbidities, medication, ejection fraction, or SMI at baseline, there may be tendencies that the subjects in the low BDNF group are older, have a lower percentage of males, a lower percentage of coronary artery disease, a higher percentage of post-operated, and a higher percentage of β blocker use (as shown in Table 1). These factors are considered potential confounding factors that should be adjusted as covariates in the multivariate analysis. Although you may not be familiar with multivariate analysis, you should use multivariate analysis (e.g., ANCOVA) to adjust for at least age and sex as covariates and statistically compare the adjusted means (and 95% confidence intervals) of the changes in AT and Peak VO2 “between” the three groups. According to the additional results obtained from the new analyses, the whole manuscript (especially, the methods section, the results section, and the discussion and conclusion sections) should be revised adequately.

Minor comments:

How was the sample size (n = 99) determined?

How was the study participants’ adherence to the cardiac rehabilitation program in the three groups?

Page 16, last line: Font size is large.

Reviewers' comments:

Reviewer's Responses to Questions

**Comments to the Author**

1. If the authors have adequately addressed your comments raised in a previous round of review and you feel that this manuscript is now acceptable for publication, you may indicate that here to bypass the “Comments to the Author” section, enter your conflict of interest statement in the “Confidential to Editor” section, and submit your "Accept" recommendation.

Reviewer #1: All comments have been addressed

Reviewer #2: All comments have been addressed

2. Is the manuscript technically sound, and do the data support the conclusions?

Reviewer #1: Yes

Reviewer #2: Yes

3. Has the statistical analysis been performed appropriately and rigorously? 

Reviewer #1: Yes

Reviewer #2: Yes

4. Have the authors made all data underlying the findings in their manuscript fully available?

Reviewer #1: Yes

Reviewer #2: Yes

5. Is the manuscript presented in an intelligible fashion and written in standard English?

Reviewer #1: Yes

Reviewer #2: Yes

6. Review Comments to the Author

Reviewer #1: (No Response)

Reviewer #2: The paper is very solid and definitely publishable. They could consider including a recent paper of close to 900 followed 7 years showing peak VO2 after CR is the strongest predictor of survival ( Carbone Set al. EHJ-QCCO 2022; 8: 361-367) and a State of the Art discussing how cardiorespiratory fitness predicts survival following CR ( Tutor A et al. Prog Cardiovasc Dis 2022; 70: 2-7.)

7. PLOS authors have the option to publish the peer review history of their article (what does this mean?). If published, this will include your full peer review and any attached files.

Reviewer #1: No

Reviewer #2: **Yes: **Carl " Chip" Lavie MD

---

## [Author Response · Author response to Decision Letter 1]

3 Oct 2023

Reviewers' comments:

Reviewer's Responses to Questions

Comments to the Author

1. If the authors have adequately addressed your comments raised in a previous round of review and you feel that this manuscript is now acceptable for publication, you may indicate that here to bypass the “Comments to the Author” section, enter your conﬂict of interest statement in the “Conﬁdential to Editor” section, and submit your "Accept" recommendation.

Reviewer #1: All comments have been addressed 

Reviewer #2: All comments have been addressed

2. Is the manuscript technically sound, and do the data support the conclusions?

The manuscript must describe a technically sound piece of scientiﬁc research with data that supports the conclusions. Experiments must have been conducted rigorously, with appropriate controls, replication, and sample sizes. The conclusions must be drawn appropriately based on the data presented.

Reviewer #1: Yes

Reviewer #2: Yes

3. Has the statistical analysis been performed appropriately and rigorously?

Reviewer #1: Yes

Reviewer #2: Yes

4. Have the authors made all data underlying the ﬁndings in their manuscript fully available?

The PLOS Data policy requires authors to make all data underlying the ﬁndings described in their manuscript fully available without restriction, with rare exception (please refer to the Data Availability Statement in the manuscript PDF ﬁle). The data should be provided as part of the manuscript or its supporting information, or deposited to a public repository. For example, in addition to summary statistics, the data points behind means, medians and variance measures should be available. If there are restrictions on publicly sharing data̶e.g. participant privacy or use of data from a third party̶those must be speciﬁed.

Reviewer #1: Yes

Reviewer #2: Yes

5. Is the manuscript presented in an intelligible fashion and written in standard English?

PLOS ONE does not copyedit accepted manuscripts, so the language in submitted articles must be clear, correct, and unambiguous. Any typographical or grammatical errors should be corrected at revision, so please note any speciﬁc errors here.

Reviewer #1: Yes

Reviewer #2: Yes

6. Review Comments to the Author

Reviewer #1: (No Response)

Reviewer #2: The paper is very solid and definitely publishable. They could consider including a recent paper of close to 900 followed 7 years showing peak VO2 after CR is the strongest predictor of survival ( Carbone Set al. EHJ-QCCO 2022; 8: 361-367) and a State of the Art discussing how cardiorespiratory fitness predicts survival following CR ( Tutor A et al. Prog Cardiovasc Dis 2022; 70: 2-7.)

Authors’ Response

Your suggestion is appreciated very much. Two manuscripts which you suggested were cited as new references #21 and #22, respectively. Also, the following sentences were added in the Discussion section.

Page 14/R2, Lines 176-178

“Recent studies have demonstrated that achieved peak VO2 after CR can serve as a predictor of long-term survival in patients with CVD [21,22].”

References

21. Carbone S, Kim Y, Kachur S, Billingsley H, Kenyon J, De Schutter A, et al. Peak oxygen consumption achieved at the end of cardiac rehabilitation predicts long-term survival in patients with coronary heart disease. Eur Hear J - Qual Care Clin Outcomes. 2022;8: 361–367. doi:10.1093/ehjqcco/qcab032

22. Tutor A, Lavie CJ, Kachur S, Dinshaw H, Milani R V. Impact of cardiorespiratory fitness on outcomes in cardiac rehabilitation. Prog Cardiovasc Dis. 2022;70: 2–7. doi:10.1016/j.pcad.2021.11.001

7. PLOS authors have the option to publish the peer review history of their article (what does this mean?). If published, this will include your full peer review and any attached ﬁles.

Do you want your identity to be public for this peer review? For information about this choice, including consent withdrawal, please see our Privacy Policy.

Reviewer #1: No

Reviewer #2: Yes: Carl " Chip" Lavie MD

---

## [Editor Report · Decision Letter 2]

16 Oct 2023

PONE-D-23-00760R2Low serum Brain-Derived Neurotrophic Factor may predict poor response to cardiac rehabilitation in patients with cardiovascular diseasePLOS ONE

Dear Dr. Tsukada,

Thank you for submitting your manuscript to PLOS ONE. After careful consideration, we feel that it has merit but does not fully meet PLOS ONE’s publication criteria as it currently stands. Therefore, we invite you to submit a revised version of the manuscript that addresses the points raised during the review process.

We look forward to receiving your revised manuscript.

Kind regards,

Yuichiro Nishida

Academic Editor

PLOS ONE

**Additional Editor Comments:**

You performed additional statistical analyses as requested by me. However, unfortunately, there are still a few severe concerns related to the additional analyses. I feel that your revision is insufficient.

The conclusion (described in the Abstract and the last paragraph of the Discussion section) is still mainly supported by the within-group comparison of AT and peak VO2. As I previously suggested, the conclusion must be based on the between-group comparison of the difference (%change) in the aerobic capacity indices.

In the Methods in the Abstract, there is no explanation of the statistical analysis mainly used. A description of the between-group comparison of the difference (%change) in aerobic capacity indices using ANCOVA is missing.

Please clearly mention which statistical analysis is mainly used to make a conclusion in the Statistical Analysis section. For instance, “Analysis of covariance (ANCOVA) followed by post-hoc analysis using Turkey’s HSD test was conducted to compare age- and gender-adjusted %increase in AT and peak VO2 after 3-week CR, as a main statistical analysis of the present study”.

In the Results section, you mentioned that “However, age- and gender-adjusted %increase in AT did not differ among the three groups”. Please show exact P values for these differences. Did age- and gender-adjusted %increase in AT not show a similar tendency as the peak VO2?

If there is not that tendency, please touch on the potential reason why the significant effect was seen only in the peak VO2 in the Discussion section.

---

## [Author Response · Author response to Decision Letter 2]

30 Nov 2023

We thank you very much that you gave us the opportunity to revise our manuscript. We greatly appreciate your insightful comments and suggestions, which have significantly contributed to enhancing the quality of our work.

The major points of the revision were as follows:

1. RE: the Conclusion 

According to your suggestion, we have revised the conclusions in the Abstract and Discussion sections to emphasized the results of the between-group comparison. 

2. RE: Methods Section in the Abstract

In the revised manuscript, we described that ANCOVA followed by post-hoc analysis using Turkey’s HSD test was conducted for between-group comparisons of age- and gender-adjusted %changes in peak VO2 and AT.

3. RE: Statistical Analysis Used for Main Results

According to your suggestion, we described in the Method section that ANCOVA followed by post-hoc analysis using Turkey’s HSD test was conducted to compare age- and gender-adjusted %increase in AT and peak VO2 after 3-week CR, as a main statistical analysis of the present study.

4. RE: Specific P Values for Age- and gender-adjusted %Increase in AT

The exact p value for age- and gender-adjusted %increase in AT was adequately added in the Abstract and Results sections.

5. RE: Discussion on the Difference between the Effects of BDNF Levels on AT and Peak VO2 

Discussion on this issue was added in the Limitations section of the Discussion.

We hope that we have successfully revised our manuscript to meet your comments and suggestion.

---

## [Editor Report · Decision Letter 3]

5 Dec 2023

PONE-D-23-00760R3Low serum Brain-Derived Neurotrophic Factor may predict poor response to cardiac rehabilitation in patients with cardiovascular diseasePLOS ONE

Dear Dr. Tsukada,

Thank you for submitting your manuscript to PLOS ONE. After careful consideration, we feel that it has merit but does not fully meet PLOS ONE’s publication criteria as it currently stands. Therefore, we invite you to submit a revised version of the manuscript that addresses the points raised during the review process.

We look forward to receiving your revised manuscript.

Kind regards,

Yuichiro Nishida

Academic Editor

PLOS ONE

Journal Requirements:

Additional Editor Comments:

The description regarding the association analysis (as one of the main statistical analyses in the present study to support the conclusion) is completely missing. Did you analyze the multivariate association of baseline BDNF (as an independent variable) with the % increases in AT or peak VO2 (as dependent variables) with adjustment for age and gender? Is the baseline BDNF (as an independent variable) treated as a continuous (mg/ml) or categorical [low, middle, high] variable? Similarly, is age (as a covariate) treated as a continuous (years) or categorical variable (another covariate gender should be treated as a categorical variable)? This important information should be clearly written in the manuscript (in the Methods in the Abstract and the Statistical Analysis in the Methods section). I have recognized that the description regarding the other main statistical analysis ANCOVA is also missing in the Methods in the Abstract.

The expression regarding the main result “correlation with baseline BDNF levels was found for the CR-induced increases in peak VO2” is somewhat strange. Please change the expression from “correlation with baseline BDNF levels was found for the CR-induced increases in peak VO2” to “ (significant) association between baseline BDNF levels and CR-induced increase in peak VO2 was found (or [significant] association of baseline BDNF levels with CR-induced increase in peak VO2 was found)” throughout the manuscript.

---

## [Author Response · Author response to Decision Letter 3]

16 Jan 2024

Additional Editor Comments:

#1. RE: The description regarding the association analysis (as one of the main statistical analyses in the present study to support the conclusion) is completely missing. Did you analyze the multivariate association of baseline BDNF (as an independent variable) with the % increases in AT or peak VO2 (as dependent variables) with adjustment for age and gender? Is the baseline BDNF (as an independent variable) treated as a continuous (mg/ml) or categorical [low, middle, high] variable? Similarly, is age (as a covariate) treated as a continuous (years) or categorical variable (another covariate gender should be treated as a categorical variable)? This important information should be clearly written in the manuscript (in the Methods in the Abstract and the Statistical Analysis in the Methods section). I have recognized that the description regarding the other main statistical analysis ANCOVA is also missing in the Methods in the Abstract.

Author’s response

We appreciate your suggestion in the decision letter to PONE-D-23-00760R1, dated on 6 Jun 2023, that “to adjust for potential confounding factors, an analysis of covariance (ANCOVA) may be used to calculate the adjusted means (and 95% confidence intervals) of the changes in AT and Peak VO2 in the three groups.” and “you should use multivariate analysis (e.g., ANCOVA) to adjust for at least age and sex as covariates.” Thus, we conducted ANCOVA for multivariate analysis as the main statistical analysis in the R2 and R3 manuscripts. But, as you pointed out, we are sorry that we failed to describe the important information regarding the ANCOVA analysis in these versions of our manuscript. Thus, in the R4 manuscript, the following changes were done as follows, to read:

In the Statistical Analysis in the Methods section,

The original sentence: “Analysis of covariance (ANCOVA) followed by post-hoc analysis using Tukey’s HSD test was conducted to compare age- and gender-adjusted %increases in AT and peak VO2 after 3-week CR, as a main statistical analysis of the present study.” (page 9/R3, lines 128-130) was changed as follows, to read:

page 9/R4, lines 130-136

“Analysis of covariance (ANCOVA) was conducted to assess the multivariate associations between baseline BDNF levels (as independent variable) and %increases in AT and peak VO2 after 3-week CR (as dependent variables) after adjustment for age and gender (as covariates), as a main statistical analysis of the present study. And, ANCOVA followed by post-hoc analysis using Tukey’s HSD test was used to compare age- and gender-adjusted %increases in AT and peak VO2 after 3-week CR. Baseline BDNF levels were treated as a categorical data based on BDNF tertiles. Age and gender were treated as a continuous and categorical data, respectively.”

In the Abstract,

New sentences were added to the Method section of the Abstract. Please note that total word count for the Abstract is limitation to be 300 words or less. Thus, we appreciate very much if you understand that it is not possible to provide more detailed information in the Abstract.

Page 2/R4, lines 32- 36

“Analysis of covariance (ANCOVA) followed by post-hoc analysis using Tukey’s HSD test was conducted to assess the multivariate associations between baseline BDNF levels categorized by BDNF tertiles (as independent variable) and %increases in AT and peak VO2 after 3-week CR (as dependent variables) after adjustment for age and gender (as covariates), as a main statistical analysis of the present study.”

And, the original words: “Analysis of covariance (ANCOVA) followed by post-hoc analysis using Tukey’s HSD test” (page 2/R3, lines 35-36) was changed to:

Page 3/R4, line 39

“ANCOVA followed by post-hoc analysis”.

After the above revisions, the total word count for the Abstract is 300 words.

#2. RE: The expression regarding the main result “correlation with baseline BDNF levels was found for the CR-induced increases in peak VO2” is somewhat strange. Please change the expression from “correlation with baseline BDNF levels was found for the CR-induced increases in peak VO2” to “ (significant) association between baseline BDNF levels and CR-induced increase in peak VO2 was found (or [significant] association of baseline BDNF levels with CR-induced increase in peak VO2 was found)” throughout the manuscript.

Author’s response

Thank you for your comment. We have rewritten the sentences, which you pointed out, throughout the manuscript, according to your suggestion.

In the Abstract, 

Page 3/R4, line 40

The original word: “correlated” (page 2/R3, line 37) was replaced to “associated”.

The original sentence: “Positive correlation with baseline BDNF levels was found for the CR-induced increases in peak VO2.” (page 3/R4, line 41) was changed as follows, to read:

Page 3/R4, lines 44-46

“A positive association between baseline BDNF levels and CR-induced increases in peak VO2 was found.”

In the Results,

Page 12/R4, line 164

The original word: “correlated” (page 13/R3, line 156) was replaced to “associated”.

The original sentence: “In contrast, significant association with baseline BDNF was not found for %increase in AT after 3-week CR was not found (p=0.1379).” (page 13/R3, lines 157-158) was changed as follows, to read:

Page 12/R4, line 165-page 13/R4, line 167

“In contrast, significant association between baseline BDNF and %increase in AT after 3-week CR was not found (p=0.1379).”

In the Discussion,

The original sentence: “a positive correlation with baseline BDNF levels was found for %increases in peak VO2, but not in AT, after 3-week CR,” (page 14/R3, lines 175-176) was rewritten as follows, to read:

Page 14/R4, line 183-184

“a positive association between baseline BDNF levels and %increases in peak VO2, but not in AT, was found after 3-week CR,”

The original sentence: “From the present study, it remains unknown why significant correlation with baseline BDNF was found for the CR-induced increase in peak VO2, but not for that in AT.” (page 19/R3, line 248-249) was changed as follows, to read:

Page 19/R4, lines 256-257

“From the present study, it remains unknown why significant association between baseline BDNF and the CR-induced increase in peak VO2, but not in AT, was found.”.

The original sentence: “a positive correlation with baseline BDNF levels was shown for the CR-induced increases in peak VO2.” (page 20/R3, line 263-264) was rewritten as follows, to read:

Page 20/R4, lines 271-272

“a positive association between baseline BDNF levels and the CR-induced increases in peak VO2 was shown.”.

Dear Editor, we are very sorry for our careless mistake. We found an incomplete sentence in the Discussion section in the R3 manuscript and have corrected it in the R4 manuscript.

In the Discussion,

The original sentence: “From this study, the association between serum BDNF and skeletal muscle strength in CVD patients.” (page 17/R3, lines 221-222) was incomplete. Thus, the following change was done in the R4 manuscript.

Page 17/R4, lines 229-230

“From this study, the association between serum BDNF and skeletal muscle strength remains undetermined in CVD patients.”

---

## [Editor Report · Decision Letter 4]

22 Jan 2024

Low serum Brain-Derived Neurotrophic Factor may predict poor response to cardiac rehabilitation in patients with cardiovascular disease

PONE-D-23-00760R4

Dear Dr. Tsukada,

We’re pleased to inform you that your manuscript has been judged scientifically suitable for publication and will be formally accepted for publication once it meets all outstanding technical requirements.

Kind regards,

Yuichiro Nishida

Academic Editor

PLOS ONE

Additional Editor Comments (optional):

Thank you for your revision. Congratulations!
---

## [Editor Report · Acceptance letter]

26 Jan 2024

PONE-D-23-00760R4 

PLOS ONE

Dear Dr. Tsukada, 

I'm pleased to inform you that your manuscript has been deemed suitable for publication in PLOS ONE. Congratulations! Your manuscript is now being handed over to our production team.

Kind regards, 

on behalf of

Dr. Yuichiro Nishida 

Academic Editor

PLOS ONE